# Transient Receptor Potential Channel Ankyrin 1: A Unique Regulator of Vascular Function

**DOI:** 10.3390/cells10051167

**Published:** 2021-05-11

**Authors:** Michael G. Alvarado, Pratish Thakore, Scott Earley

**Affiliations:** Center for Molecular and Cellular Signaling in the Cardiovascular System, Department of Pharmacology, Reno School of Medicine, University of Nevada, Reno, NV 89557-0318, USA; michaelalvarado@med.unr.edu (M.G.A.); pthakore@med.unr.edu (P.T.)

**Keywords:** TRPA1, endothelial cell, neurovascular coupling, inflammation, stroke

## Abstract

TRPA1 (transient receptor potential ankyrin 1), the lone member of the mammalian ankyrin TRP subfamily, is a Ca^2+^-permeable, non-selective cation channel. TRPA1 channels are localized to the plasma membranes of various cells types, including sensory neurons and vascular endothelial cells. The channel is endogenously activated by byproducts of reactive oxygen species, such as 4-hydroxy-2-noneal, as well as aromatic, dietary molecules including allyl isothiocyanate, a derivative of mustard oil. Several studies have implicated TRPA1 as a regulator of vascular tone that acts through distinct mechanisms. First, TRPA1 on adventitial sensory nerve fibers mediates neurogenic vasodilation by stimulating the release of the vasodilator, calcitonin gene-related peptide. Second, TRPA1 is expressed in the endothelium of the cerebral vasculature, but not in other vascular beds, and its activation results in localized Ca^2+^ signals that drive endothelium-dependent vasodilation. Finally, TRPA1 is functionally present on brain capillary endothelial cells, where its activation orchestrates a unique biphasic propagation mechanism that dilates upstream arterioles. This response is vital for neurovascular coupling and functional hyperemia in the brain. This review provides a brief overview of the biophysical and pharmacological properties of TRPA1 and discusses the importance of the channel in vascular control and pathophysiology.

## 1. A Brief Introduction to TRP Channels

The transient receptor potential (TRP) superfamily is a group of cation-permeable channels, the members of which reside on the plasma membranes and endomembranes of excitable and non-excitable cells throughout the body [1]. TRP channels mediate the transmembrane flux of monovalent (K^+^ and Na^+^) and divalent (Ca2+ and Mg2+) cations down their physiological electrochemical gradients, contributing to the regulation of membrane potential, Ca2+-dependent signaling cascades, and other critical processes [2]. The members of the TRP superfamily share evolutionarily conserved elements of genetic and molecular architecture but have evolved distinct gating mechanisms and sensitivities to chemical, osmotic, and thermal sensory stimuli [2,3]. Since the initial discovery and characterization of the *trp* locus in *Drosophila* phototransduction mutants in 1969 [4], the TRP channel collective has become broadly recognized for the complex ability to detect, integrate, and transduce a multitude of physiologically distinct polymodal triggers to control physiological processes ranging from axon guidance to vascular function [5,6,7,8,9,10,11].

Nine distinct TRP channel subfamilies have been identified, of which six are present in mammals [12]. The six mammalian TRP subfamilies are composed of 28 genes encoding TRP subunit proteins. TRP channels share a common topological architecture based on the tetrameric assembly of six-transmembrane α-helix domain subunits to form a functional cation-permeable pore—an archetypal feature commonly found in a variety of ion channel families [3]. Tetrameric assembly of TRP subunits gives rise to functional homomeric and heteromeric cation channels [3]. Several TRP channel subunits are characterized by domains composed of repeated 33-amino-acid–residue sequences that form an anti-parallel helix-turn-helix motif known as an ankyrin repeat domain. Repeated ankyrin domains are thought to function as intracellular molecular adapters [2,13] and enable mechanical elasticity [14,15]; in the case of TRP channels, they also modulate channel assembly and activity [16,17,18]. The lone member of the mammalian ankyrin (A) TRP subfamily, TRPA1, is distinguished by its extensive and eponymous ankyrin repeat domain [19].

This review focuses on the role of TRPA1 channels in the vasculature. The biophysical properties of TRPA1 channels are briefly described, followed by an overview of our current understanding of TRPA1 channels in vascular regulation. We also discuss empirical evidence spotlighting TRPA1 as an initiator of neurovascular coupling (NVC), the process that nourishes metabolically active neurons in the brain in real time. Finally, we discuss the role of TRPA1 in several types of vascular pathology.

## 2. The Mammalian TRPA1 Channel: One of a Kind

The human *TRPA1* gene encoding the TRPA1 subunit is present on chromosome 8, and the mouse *Trpa1* gene is present on chromosome 1. The general structure of TRPA1 polypeptide subunits, predicted from cloned *TRPA1,* is characterized by a six-transmembrane pore-forming domain shared with other superfamily members and numerous ankyrin repeats at the cytosolic amino (N)-terminus [20]. TRPA1 subunits are 1119 amino acids long, forming a ~127 kDa protein with a small extracellular domain and intracellular N- and carboxyl (C)-termini, which collectively account for nearly 80% of the total protein mass [21]. The molecular architecture of TRPA1 was solved at near-atomic resolution (~4Å) using single-particle electron cryo-microscopy (cryo-EM) by Paulsen et al., revealing details of its Ca^2+^-conducting pore, C-terminal coiled-coil region, and extensive N-terminal ankyrin domain [22] (Figure 1). Sixteen ankyrin repeat domains are present in the cysteine-rich intracellular N-terminus of human TRPA1 subunits, and 14 ankyrin repeats are present in mouse TRPA1 subunits [23]. The 12th ankyrin repeat on TRPA1 is thought to form an EF-hand-like Ca2+-sensing domain responsible for intracellular channel modulation [19,24]. Ca^2+^-dependent modulation is believed to be conferred by residues D466, L474 and D477 found within the EF-hand domain [17,24,25,26], and simulation studies have shown that it induces a local conformational shift that stiffens the ankyrin repeats, causing the TRPA1 pore to open [27]. Additionally, TRPA1 is endowed with a conserved Ca2+-binding motif (residues A1080–A1082) [28] and a voltage-sensing domain (residues R975, K988, and K989) [29] in the C-terminus. A negatively charged selectivity filter comprising residues E920, E924, and E930 is located at the outer pore entrance [30]. This pore entrance-residing negatively charged filter attracts cations and repels anions [30]. Once past the selectivity filter, cations encounter a dually restrictive permeation pathway formed by the tetrameric subunit assembly. The first site is a negatively charged ring of residues (D915 from each subunit) located below the pore entrance that enhances cation attraction while maintaining a sufficiently large diameter (7Å) for the passage of Ca2+ [30]. Residues I957 and V961 form the second permeation site, with its slightly smaller pore (6Å) possibly indicating the channel’s closed state [22,30]. Agonist stimulation of TRPA1 results in progressive, but reversible, changes in the selectivity filter that cause dilation of the pore, allowing the passage of large molecules (up to 10Å in diameter) [31,32,33] such as the cationic dye Yo-Pro [33]. Intriguingly, whether permeation of large molecules is solely attributable to changes in channel permeability or if fluctuating intracellular ion concentration dynamics play a role is currently unclear [34].

TRPA1 channels are permeable to monovalent and divalent cations. However, TRPA1 is more permeable to Ca^2+^ ions than monovalent cations (P_Ca_:P_Na_:P_K_ ≈ 7.9:1.0:0.98) [32,35,36], with the Ca^2+^ current constituting approximately 20% of the total mixed cation current [37]. Under physiological conditions, the unitary conductance of TRPA1 in the inward and outward direction is ~70 and 100 pS, respectively [35,36]. Interestingly, both intracellular and extracellular Ca^2+^ modulate TRPA1 activity, with increased intracellular Ca^2+^ concentration produced by Ca^2+^ influx potentiating agonist-induced TRPA1 currents and facilitating their initiation [38], as well as with increased extracellular Ca^2+^, similarly potentiating TRPA1 currents stimulated by electrophilic and non-electrophilic agonists [39].

## 3. Pharmacology of TRPA1 Channels

### 3.1. Activators

TRPA1 can be activated by plant-derived pungent compounds, environmental irritants, local anesthetics, synthetic compounds, and reactive oxygen species (ROS) [1,38,41,42,43,44,45,46] (Table 1). TRPA1 agonists can be classified into two types: covalently modifying electrophilic compounds, such as allyl isothiocyanate (AITC), allicin, cinnamaldehyde and JT010 [38,47,48], that increase the probability of channel opening and non-covalently modifying, non-electrophilic compounds, such as menthol and delta-9-tetrahydrocannabinol (THC) [49,50]. TRPA1 activation by electrophilic compounds occurs at thiol groups of cysteine residues in the N-terminal domain, which crystal structure analyses have suggested form a ligand-binding pocket [17,21,22,27,51,52]. Currently, only five N-terminal cysteine residues (human: C414, C421, C621, C641, and C665; mouse: C415, C422, C622, C642, and C666) are recognized as being involved in electrophile-stimulated covalent modification of TRPA1, with residue C621 being the most reactive [53,54]. Further, the extensive ankyrin repeat domains on TRPA1 confer the evolutionarily advantageous ability to sense naturally pungent and potentially toxic electrophilic compounds [55]. Specific ankyrin repeats in the N-terminal domain are thought to be involved in TRPA1 activation. For example, ankyrin repeats 3–8 and 10–15 in TRPA1 from *Crotalus atrox* (western diamondback rattlesnake) are activated by infrared heat [56], whereas ankyrin repeats 11–16 in human (11–15 in mouse) TRPA1 are instead triggered by chemical irritants and inflammatory agents [54,57], including the dietary compounds allicin (garlic), AITC (mustard oil) [38,47] and cinnamaldehyde [47].

A non-electrophilic, cell-penetrating peptidergic toxin (WaTx) from *Urodacus manicatus* (Australian black rock scorpion) was recently shown to interact with an intracellular electrophile ligand-binding domain involving residues C621 and C641 [65]. This interaction effectively stabilized TRPA1 in a prolonged open conformation but diminished Ca2+ permeability. Remarkably, this mechanism of action was distinct in that WaTx is a non-covalent gating modifier that does not directly promote the TRPA1 open state but instead stabilizes it. In contrast, another non-electrophilic compound, GNE551, was found to bind a hydrophobic transmembrane binding site (residue Q940) of TRPA1 to increase the channel’s open state without affecting Ca^2+^ permeability [64]. Considering that most non-electrophilic TRPA1 agents have generally been regarded as non-selective compounds with low potency and efficacy [38,43], the continued discovery and development of highly selective non-covalent compounds present novel strategies for deciphering the properties and physiological significance of non-covalent TRPA1 activation.

### 3.2. Blockers

Early studies on TRPA1 used non-selective inhibitors such as camphor [66] and ruthenium red [25] to block TRPA1 activity, but these agents are no longer used experimentally [67]. Instead, more recently developed selective pharmacological inhibitors of TRPA1 are employed to dissect the channel’s physiological functions (Table 2). Among these, HC-030031 is the most frequently used TRPA1 blocker [68]. Chembridge-5861528, a derivative of HC-030031, has similar potency and specificity with improved solubility [69]. AP-18 is a partial agonist that exerts an inhibitory effect by desensitizing the channel [70,71]. A-967079, Compound 10, and Compound 31 are the most potent TRPA1 inhibitors [72,73,74].

### 3.3. Endogenous Regulators

TRPA1 is endogenously activated by ROS and ROS metabolites [52], such as the lipid peroxidation metabolites 4-hydroxy-2-noneal (4-HNE) [46,61] and the related 4-oxononeal (4-ONE) and 4-hydroxyhexenal (4-HHE) [62]. Such reactive electrophiles activate the channel by covalently and reversibly modifying N-terminal cysteine and lysine residues [17,55,76]. It has also been reported that TRPA1 is activated by H_2_O_2_ and the cyclopentenone prostaglandin 15-deoxy-delta(12,14)-prostaglandin J(2) [15d-PGJ(2)] [41], the hydroxyl radical (OH•) [77], nitric oxide (NO) and peroxynitrite [78], as well as nitroxyl (HNO) [79]. In addition to sensing endogenous oxidative stress-related substances, it has been proposed that TRPA1 senses molecular oxygen (O_2_) [76]. Interestingly, high-resolution cryo-EM has revealed the presence of additional cysteine and lysine residues (C727, K771, and C834) within the TRPA1 transmembrane core that are predicted to be exposed to the lipid environment, where they may interact with lipophilic electrophiles [22]. This is a developing area of investigation.

## 4. Regulation of Vascular Tone by TRPA1 Channels

### 4.1. TRPA1 Channels and Neurogenic Vasodilation

In a pioneering study, Bautista et al. demonstrated that exogenous application of the TRPA1 agonist AITC, allicin, or diallyl disulfide (DADS) to precontracted rat mesenteric arteries induced relaxation [58]. This response was blocked by capsaicin-induced sensory nerve denervation, pre-treatment of the tissue with ruthenium red, or exposure to the calcitonin gene-related peptide (CGRP) receptor antagonist CGRP_8-37_. Application of the TRP vanilloid 1 (TRPV1) blocker capsazepine had no effect. This study also showed that TRPA1 is expressed in a subgroup of perivascular sensory neurons that innervate mesenteric arteries—neurons that also contain the potent vasodilator CGRP [80]. Bautista et al. concluded that subpopulations of sensory nerve terminals express TRPA1 channels that, when activated, cause the release of CGRP and induce dilation of the adjacent vasculature (Figure 2).

The demonstration that cinnamaldehyde induces dilation of isolated mouse mesenteric arteries in vitro provided further evidence of TRPA1-mediated vascular control [81]. Critically, this study demonstrated that cinnamaldehyde-induced dilation of mesenteric arteries was significantly attenuated in global *Trpa1*-knockout (*Trpa1*-KO) mice. Additionally, topical application of cinnamaldehyde or AITC to the ears of anesthetized mice in vivo caused an increase in blood flow in wild-type mice, but not in *Trpa1*-KO mice [81]. A follow-up study showed that TRPA1-mediated increases in blood flow were diminished following application of the TRPA1 blocker HC-030031 or CGRP_8-37_ and were blunted in global CGRP-knockout (*Calca*-KO) mice [82]. Moreover, intraplantar injection of 4-ONE into the hind paws of anesthetized wild-type mice resulted in local increases in blood flow, an effect that was absent in both *Trpa1*- and *Calca*-KO mice but was unaltered in global *Trpv1*-KO mice [83]. In vitro work further unveiled a link between TRPA1 and meningeal vasodilation by demonstrating that the release of CGRP by cultured rat trigeminal neurons was driven by exposure to AITC, acrolein or cinnamaldehyde, and could be attenuated by HC-030031 [84]. Notably, this study demonstrated transient increases in meningeal blood flow following nasally administered AITC or acrolein. These effects were inhibited by HC-030031 and CGRP_8-37_, providing strong evidence of a link between TRPA1 and CGRP in regulating vascular tone. In line with this, Eberhardt et al. proposed that vascular tone is regulated by the gasotransmitter hydrogen sulfide (H_2_S), which transforms endogenous NO to HNO and activates the neuroendocrine HNO–TRPA1–CGRP signaling pathway [79]. In this study, laser Doppler recordings of rat meningeal blood flow demonstrated flow increase following topical application of H_2_S. Increased blood flow was diminished following topical application of HC-030031 or by intravenous injection of the nitric oxide synthase (NOS) inhibitor L-NMMA [79]. Additionally, in the rat medullary brainstem, intravenous administration of Na_2_S, a fast-releasing H_2_S donor, led to increased blood flow and elevated CGRP in the cerebrospinal fluid. Brainstem blood flow was attenuated following application of CGRP antagonists CGRP_8-37_ and BIBN 4096BS, TRPA1 antagonist HC-030031, or the NOS inhibitor L-NAME. Notably, circulating CGRP was significantly diminished in the presence of HC-030031 and L-NAME confirming that an interaction between H_2_S and NO, in which NO is reduced to HNO, is necessary for activating the HNO–TRPA1–CGRP pathway [79]. This group also demonstrated this mode of action in isolated mouse mesenteric arteries, further supporting their hypothesis of vascular tone regulation by CGRP produced following TRPA1 channel activation by HNO. Collectively, these data convincingly supported the hypothesis put forth by Bautista et al. that TRPA1 channels are present on sensory neurons of the vascular adventitia and, upon stimulation, initiated the Ca^2+^-dependent release of the neuropeptide CGRP onto adjacent vascular smooth muscle cells (VSMCs) [58]. CGRP binds CGRP receptors on VSMCs resulting in membrane hyperpolarization [85], VSMC relaxation and, ultimately, vasodilation (Figure 2).

A few studies have reported that ROS and NO are produced upon activation of TRPA1 channels in sensory nerves and cause vasodilation. In one study, cinnamaldehyde-induced increases in blood flow in the mouse ear were attenuated by scavenging ROS or inhibiting neuronal NOS (nNOS) using S-methyl-L-thiocitrulline (SMTC) [82]. Superoxide and NO react to form peroxynitrite, a highly reactive, short-lived oxidant species [86] capable of mediating vasodilation in a cGMP-dependent manner [87,88]. It was further found that cinnamaldehyde-induced vasodilation in mice was diminished following treatment with the peroxynitrite scavenger FeTPPs [82]. How superoxide is produced following TRPA1 activation remains unknown and warrants further investigation.

Peripheral blood vessels constrict in response to cold temperatures, followed by a vasodilatory recovery phase [89,90]. The mechanisms underlying this biphasic vascular response are poorly understood. Based on data indicating that TRPA1 channels expressed in mammalian cell lines are activated by temperatures in the range of 5–18 °C [67,91,92], it was proposed that TRPA1 channels could mediate vascular responses to noxious cold [93]. Aubdool et al. demonstrated that exposing the hind paw of wild-type mice to 10 °C water resulted in a biphasic vascular response which was absent in *Trpa1*-KO mice and wild-type animals treated with HC-030031 [93]. The authors of this study proposed that the initial vasoconstriction was due to TRPA1-dependent generation of ROS, which induced the release of sympathetic neuron-derived norepinephrine. However, the precise mechanism by which ROS is produced downstream of TRPA1 activation was not determined. Denervation of sympathetic nerves with guanethidine or phentolamine-induced blockade of α-adrenoceptors only partially reduced vasoconstriction, suggesting the involvement of additional contractile pathways [93]. This study demonstrated that TRPA1 is involved in the cold-induced vasodilatory phase, as evidenced by the ability of intravenously administered HC-030031 to block cold-induced vasodilation following peak contraction. That sensory nerves mediate this vasodilatory phase is supported by the observation that vasodilation was diminished by inhibition of CGRP or nNOS or by sensory nerve denervation. However, the role of TRPA1 channels as a cold sensor is controversial, with several reports suggesting the TRPA1 channels are not activated by cold [38,94,95]. Conclusive evidence supporting the role of mammalian TRPA1 channels as primary cold sensors in the vasculature has yet to be reported.

### 4.2. TRPA1 and Endothelium-Dependent Vasodilation

Studies from our laboratory and those of others have demonstrated that TRPA1 channels are present in the endothelium of cerebral pial arteries and parenchymal arterioles [96,97,98,99] but are not found in the endothelium of arteries from other vascular beds, including the mesentery, heart, kidney, and dermis [100]. Notably, expression of TRPA1 has not been detected in VSMCs. Activation of TRPA1 channels in cerebral arteries and arterioles with AITC or 4-HNE causes endothelium-dependent vasodilation, a response that is blunted by the selective TRPA1 inhibitor HC-030031 and is absent in endothelial cell-specific TRPA1-KO (*Trpa1*-ecKO) mice [100]. This vasodilator response is insensitive to the inhibition of NO synthase and cyclooxygenase activity but is abolished by blockade of small- and intermediate-conductance Ca^2+^-activated K^+^ (SK and IK, respectively) channels [100]. Immunolabeling studies have demonstrated that TRPA1 channels are enriched within specialized regions of the endothelial cell plasma membrane that penetrate the internal elastic lamina and interface with underlying VSMCs [96,100]. These structures, known as myoendothelial projections (MEPs) [96], house vital signaling complexes that modulate the membrane potential and contractility of adjacent VSMCs. IK channels and inositol triphosphate receptors (IP_3_Rs) on the endoplasmic reticulum (ER) are also enriched in the MEPs of cerebral pial arteries, and these proteins form signaling complexes with TRPA1 channels [96]. An interesting study from Qian et al. demonstrated that Ca^2+^ influx through TRPA1 channels stimulated by AITC triggered Ca^2+^-induced-Ca^2+^ release from proximal IP_3_Rs to generate transient, high-amplitude Ca^2+^ signals that are localized to MEPs [97]. These data support the concept that Ca^2+^ influx through TRPA1 channels stimulates further Ca^2+^ release from the ER via IP_3_R to generate subcellular microdomains of high [Ca^2+^] that stimulate IK and SK channel activity. The ensuing efflux of K^+^ hyperpolarizes the endothelial cell plasma membrane. This signal is electrically propagated to the overlying VSMCs through myoendothelial gap junctions (MEGJ) to cause hyperpolarization and relaxation (Figure 3).

The endogenous regulation of TRPA1 channels in the cerebral endothelium has been investigated by recording Ca^2+^ influx through single TRPA1 channels known as “TRPA1 sparklets” using innovative “optical patch-clamp” techniques employing total internal reflection fluorescence (TIRF) microscopy [100,101]. Using this technique, exogenous application of 4-HNE was observed to increase TRPA1 sparklet frequency and relax cerebral arteries isolated from wild-type mice [100]. This response was blocked by HC-030031 and was absent in tissues isolated from *Trpa1*-ecKO mice, demonstrating that 4-HNE activates TRPA1 channels in the intact cerebral endothelium. Interestingly, TRPA1 was shown to colocalize with the ROS-generating enzyme NADPH oxidase 2 (NOX2) within MEPs of cerebral arteries, and exogenous application of the NOX2 substrate NADPH increased TRPA1 sparklet frequency and caused vasodilation [100]. These responses were blunted by inhibiting NOX activity, degrading extracellular H_2_O_2_ with catalase, or chelating iron to block the formation of OH• via the Fenton reaction, thereby diminishing lipid peroxidation and the production of 4-HNE [102]. These data collectively provide evidence that endogenously generated NOX2-derived lipid peroxidation products activate TRPA1 channels to cause endothelium-dependent dilation of cerebral arteries (Figure 3). These findings further suggest that TRPA1 channels can act as ROS sensors within the cerebral microvasculature and may provide adaptability during redox status changes and oxidative stress in the brain.

### 4.3. TRPA1 and Neurovascular Coupling

Neurons within the brain have a limited ability to store energy and the substances required for its generation. Therefore, blood flow must be precisely and rapidly redirected to metabolically active regions through a process known as neurovascular coupling (NVC) [103,104,105,106,107]. The brain’s vascular network consists of an extensive arborescent complex of surface pial arteries, penetrating parenchymal arterioles, and a vast network of capillaries (Figure 4). The traditional functions of cerebral capillaries, which are in intimate contact with neurons in the brain, are to regulate the exchange of gases, nutrients, and waste with the brain parenchyma and maintain the integrity of the blood–brain barrier. However, a paradigm-shifting study by Longden et al. demonstrated that brain capillaries function as a sensory web capable of detecting metabolic activity in the brain and orchestrating the re-direction of blood flow to these regions [108]. This group showed that extracellular K^+^ released by active neurons triggers inwardly rectifying K^+^ (K_ir_) channels on cerebral capillary endothelial cells, generating a hyperpolarizing signal that propagates through the vasculature to upstream parenchymal arterioles to cause dilation and a subsequent increase in local blood perfusion at the signal source [108,109]. These data demonstrate a central role for cerebral capillaries in NVC.

Reasoning that NVC is essential for brain health and function, our team proposed that overlapping and redundant neuronal activity sensors are present in brain capillary endothelial cells [110]. Using whole-cell patch-clamp electrophysiology, we determined that TRPA1 channels are present and functional on native brain capillary endothelial cells. In a series of ex vivo experiments utilizing a recently characterized isolated ex vivo microvascular preparation that allows for pressure myography studies of brain arteriole segments attached to their capillary branches [108], we found that direct application of the TRPA1 agonists AITC or 4-HNE onto distal capillary extremities elicited transient, reversible, and reproducible dilatation of the connected upstream parenchymal arteriole [110]. This response was diminished in preparations from *Trpa1*-ecKO mice, or by superfusing the TRPA1 antagonist HC-030031 or severing the capillaries from their upstream arteriole in preparations from wild-type mice. Measurements of changes in red blood cell flux at the capillary level in vivo using two-photon laser-scanning microscopy yielded similar results [110]. Focal application of AITC directly onto a single capillary increased both red blood cell flux within the stimulated capillary and the cross-sectional area of upstream arterioles, indicating a conducted vasodilatory response [110]. These increases in capillary red blood cell flux and arteriole cross-sectional area were absent in *Trpa1*-ecKO mice. These data collectively support the concept that TRPA1 is functionally expressed on brain capillary endothelial cells, and that their activation produces a vasodilatory signal that propagates retrogradely towards upstream arterioles.

The physiological relevance of TRPA1 in brain capillaries was demonstrated by assessing functional hyperemia in the somatosensory cortex of mice in vivo using laser Doppler flowmetry [111,112]. Stimulating contralateral whiskers for 5 s produced a reproducible increase in blood flow that was significantly diminished by blocking TRPA1 with HC-030031 and was blunted in *Trpa1*-ecKO mice [110]. In contrast, blood flow increases in response to short-duration (1 and 2 s) stimulation were unchanged in mice treated with HC-030031 and did not differ between wild-type and *Trpa1*-ecKO mice [110]. Thus, it appears that endothelial cell TRPA1 channels are not involved in functional hyperemic responses to short-duration (<5 s) stimuli, but are vital for maintaining responses during more prolonged stimulation.

TRPA1 channels in brain capillary endothelial cells initiate a novel biphasic propagating vasodilator signal [110] (Figure 5). An analysis of the conduction kinetics of vasodilator signals revealed that the propagation velocity of vasodilatory signals was significantly slower following application of TRPA1 agonists onto capillaries compared with that following application of K^+^. A further comparison of different vascular segments confirmed these results, showing that signals initiated by TRPA1 channels propagate much more slowly through deeper regions of the capillary network than those initiated by K_ir_ channel activation [110]. However, we also showed that, in the post-arteriole transitional segment—a mural cell-rich region that bridges deeper reaches of the capillary network with its feeding parenchymal arteriole [113]—signal propagation is rapid and comparable to that produced by activating K_ir_ channels [108,110]. This analysis supports the concept that TRPA1-dependent propagation occurs through a biphasic mechanism in which activation of TRPA1 produces an initial signal that travels slowly through the distal capillary network and is converted into a rapid electrical signal at the transitional segment [110]. The relevant mechanisms were investigated by focally applying AITC onto capillaries in microvascular preparations from mice expressing the genetically encoded Ca^2+^ indicator GCaMP8 exclusively in endothelial cells (*Cdh5*-GCaMP8 mice) [114]. This maneuver produced a slowly propagating, short-range (3–5 cells) intercellular Ca^2+^ signal that was blocked by HC-030031 [110]. Notably, propagation of Ca^2+^ signals through the capillary bed and conducted vasodilatory responses initiated by activation of TRPA1 channels in capillaries were blocked by inhibiting P_2_X purinergic receptors, degrading extracellular ATP with apyrase, or pharmacologically inhibiting pannexin-1 (Panx-1) channels [110]. Consistent with this latter observation, propagative vasodilation induced by AITC was attenuated in preparations from *Panx1*-ecKO mice. These data suggest that activation of TRPA1 channels in endothelial cells deeper in the capillary bed causes Panx-1 channel-mediated release of ATP, which acts on P_2_X receptors in neighboring cells to generate slowly propagating intercellular Ca^2+^ signals in the brain capillary endothelium [110] (Figure 5). Propagating intercellular Ca^2+^ signals initiated by TRPA1 in distal capillaries are converted to fast-propagating electrical signals by SK and IK channels upon reaching the post-arteriole transitional segment. These fast electrical signals require K_ir_ channels activity and ultimately dilate the upstream vasculature.

## 5. TRPA1 Channels and Vascular Disease

### 5.1. Inflammation

The vascular component of neurogenic inflammation is thought to be triggered by neuronal TRPA1-mediated release of CGRP, substance P, and other proinflammatory vasoactive peptides. Trevisani et al. first demonstrated that intraplantar application of the TRPA1 agonist 4-HNE to the hind paw of rats elicited edema formation and pain-associated behavior [46]. In a model of cigarette smoke inhalation-induced inflammation, Andre et al. further demonstrated that tracheal plasma extravasation was attenuated by blocking TRPA1 channel activity with HC-030031 [115]. Moreover, this study reported increased capillary permeability in wild-type mice, but not global *Trpa1*-KO mice, treated with an aqueous extract of cigarette smoke, suggesting a TRPA1-mediated diminution of the endothelial cell-to-cell barrier. In separate studies, TRPA1 was found to mediate neurogenic inflammation following stimulation with N-acetyl-p-benzoquinone imine (NAPQI), a toxic byproduct of acetaminophen, and the bacterial endotoxin lipopolysaccharide, among other stimuli [116,117]. Interestingly, it was reported by Graepel et al. that, although TRPA1 channel activity mediates the vasodilatory effect of neurogenic inflammation, it is not responsible for edema formation in the mouse hind paw [83]. No conclusive explanation for the inconsistency in edema formation currently exists, but these observations may collectively indicate that TRPA1 channel activity mediates neurogenic inflammation in a tissue-specific manner. Taken together, the reported findings suggest that the TRPA1 channel may serve as a novel therapeutic target for treatment of inflammatory pathological states of increased capillary permeability and inflammation-associated edema.

It was suggested by Zhao et al. that TRPA1 channel signaling confers protection against the development of atherosclerosis [118], the leading cause of vascular disease worldwide [119]. Atherosclerosis is a progressive and chronic inflammatory condition associated with accretion of fatty plaques within the innermost layer of arteries that eventually disrupts blood flow. Over time, this disruption can compound and manifest as an aneurysm or stroke. In apolipoprotein E (*Apoe*)-KO mice, a mouse model of atherosclerosis, TRPA1 channel activation with AITC was found to suppress atherosclerotic progression [118,120,121]. Pharmacological blockade or genetic deletion of TRPA1 in *Apoe*-KO mice resulted in aggravated atherosclerotic lesions, hyperlipidemia, and increased circulatory inflammatory cytokines [118]. Macrophages, the central synthesizers of inflammatory and metabolic signals in atherosclerotic lesions [122], express TRPA1 channels. As active participants in atherosclerosis, macrophages ingest lipoproteins and generate foam cells that are retained within and contribute to atherosclerotic plaque growth [123]. The inability of these macrophages to migrate aggravates the local proinflammatory atherosclerotic environment. Zhao et al. proposed blocking TRPA1 channel activation-dependent production of TNF-α (tumor necrosis factor-α) as a strategy for reducing lipoprotein ingestion and atherosclerotic plaque formation [118]. However, the link between TRPA1, macrophages, and atherosclerosis remains poorly understood. Thus, future studies will be necessary to elucidate the role of TRPA1 channel signaling in macrophage recruitment and polarization under conditions of atherosclerosis.

Inflammation is a critical component of chronic kidney disease (CKD), a syndrome that persistently compromises kidney structure and function [124]. Factors including oxidative stress, acidosis, and the increased production (and decreased clearance) of proinflammatory cytokines contribute to the chronic inflammation in CKD [125]. Despite advances in the management of CKD, this syndrome continues to be associated with unacceptable levels of morbidity and mortality [125]. A clinical observational study revealed that TRPA1 is upregulated in the renal tubules of patients with acute kidney injury (AKI) [126], leading study authors to suggest that TRPA1 is associated with the progression of AKI to CKD. Methylglyoxal, a reactive glucose metabolite, is elevated in CKD [127] and may be a central contributor to the inflammatory, structural, metabolic, and hemodynamic pathology of CKD [128]. Eberhardt et al. reported that methylglyoxal facilitates Ca^2+^ influx via TRPA1 channels in primary sensory neurons and transfected cells and promotes the release of CGRP from peptidergic nerves [129]. Methylglyoxal was also shown to activate TRPA1 channels by modifying the N-terminal residues, K710, C621, C641 and C665 (although additional unresolved cysteine residues are implicated in the response) [129]. Andersson et al. reported that methylglyoxal evokes acute pain and long-lasting neuropathy in wild-type mice, but not *Trpa1*-KO mice, providing in vivo evidence for methylglyoxal-mediated TRPA1 channel activation [130]. Functional studies linking TRPA1 channel activity with kidney physiology are lacking. However, considering that TRPA1 channels can be activated by methylglyoxal, a suspected contributor to CKD progression, it would be interesting to determine if TRPA1 is a suitable therapeutic target for CKD treatment.

### 5.2. Hypertension

Cardiovascular morbidities such as coronary heart disease, heart failure, and stroke converge on elevated systolic or diastolic blood pressure [131]. This hypertensive response is aggravated in mice lacking CGRP, a vasoactive neuropeptide released upon TRPA1 channel activation [132,133]. Bodkin et al. demonstrated that, although TRPA1 channel signaling was not involved in mediating the vascular component of elevated blood pressure, it might be involved in the inflammatory component of hypertension [134]. This was determined using an angiotensin II (Ang II) model of hypertension, in which the blood pressure elevation induced by 14 days of subcutaneous Ang II infusion was indistinguishable between wild-type and *Trpa1*-KO mice. Notably, the proinflammatory cytokine interleukin-6 was diminished in hypertensive *Trpa1*-KO mice [134]. In kidney studies, Ma et al. similarly reported that systolic blood pressure was indistinguishable between wild-type and *Trpa1*-KO mice following Ang II subcutaneous infusion for 28 days [135]. The resulting hypertensive *Trpa1*-KO mice also exhibited severe kidney dysfunction and elevated blood urea nitrogen, serum creatinine, renal fibrosis, and renal inflammatory cytokine levels [135]. Collectively, these observations have led to the proposal that TRPA1 channel signaling confers protection against kidney dysfunction by inhibiting the production of proinflammatory mediators. Thus, TRPA1-directed therapeutics may not ameliorate the vascular pathology of hypertensive blood pressure but may instead protect against the inflammatory component of the pathology.

### 5.3. Stroke

Cerebral tissue damage due to stroke is the second-leading cause of death worldwide, and half of all stroke survivors are afflicted with chronic disabilities [136]. Tissue hypoxia arises as a consequence of blockages within the cerebral vasculature, leading to ischemic stroke. Cerebral arteries dilate in response to hypoxic conditions, a response that serves to improve oxygen delivery to ischemic brain regions [137]; however, the mechanisms underlying this adaptive response remain poorly understood. Our laboratory investigated the concept that hypoxia dilates cerebral arteries by activating TRPA1 channels in the endothelium. To recapitulate relevant pathophysiological pO_2_ levels consistent with ischemia-induced peri-infarct pO_2_ [138,139], we investigated the effects of a pO_2_ of ~10-15 mmHg on TRPA1 activity [138]. We found that hypoxic conditions increased 4-HNE levels and TRPA1 sparklet frequency in the cerebral endothelium. Consistent with this, acute hypoxia induced dilation of cerebral arteries and penetrating parenchymal arterioles in wild-type but not *Trpa1*-ecKO mice [138]. The membrane-permeant mitochondrial-targeting antioxidant mitoTEMPO diminished hypoxia-induced dilation, suggesting that the generation of ROS by mitochondria under hypoxic conditions served to activate TRPA1 and cause vasodilation [138]. In vivo experiments using a middle cerebral artery occlusion model of ischemic stroke showed that infarct areas produced in *Trpa1*-ecKO mice were substantially larger than those in wild-type controls, indicating that hypoxia-induced TRPA1 channel activation ameliorates pathological changes in cerebral tissue following ischemic insult [138]. Notably, shortly following occlusion, infarct areas were reduced in control mice treated with the TRPA1 agonist cinnamaldehyde, further supporting the concept that increased blood flow due to endothelial TRPA1-dependent vasodilation protects against hypoxia-associated brain damage. Collectively, these data indicate that TRPA1 channels in the cerebral endothelium are vital sensors of acute hypoxia and mediate protective vasodilation in brain areas affected by ischemia.

## 6. Summary and Conclusions

TRPA1 channels are distinguished structurally by their extensive N-terminal ankyrin repeat domains and functionally by their sensitivity to naturally pungent and potentially toxic electrophilic compounds. Activation of TRPA1 channels elicits vasodilation through several distinct signaling pathways. Stimulation of TRPA1 channels on sensory neurons surrounding blood vessels causes release of the potent vasodilator CGRP. Active TRPA1 channels on endothelial cells of cerebral arteries and arterioles engage Ca^2+^-activated K^+^ channels to hyperpolarize and relax VSMCs. Stimulation of TRPA1 channels on brain capillary endothelial cells orchestrates a propagating response that dilates upstream parenchymal arterioles. These TRPA1-mediated processes are vital for neurogenic- and endothelium-dependent vasodilation, NVC and functional hyperemia in the brain, as well as adaptability to hypoxia and oxidative stress. Further studies are necessary to further elucidate the endogenous regulators of TRPA1 channels under physiological and pathophysiological conditions. It will also be important to investigate the significance of TRPA1 channels in other types of cells that contribute to vascular function. For example, astrocytes express functional TRPA1 channels [140,141,142,143], and astrocytic endfeet enveloping capillaries and neighboring vascular segments are optimally positioned to release vasoactive signaling molecules [144,145,146]; however, nothing is known about how TRPA1 channel activity in astrocytes influences vascular function. Also of considerable interest are future preclinical studies investigating how pharmacological manipulation of TRPA1 channels could be used to treat inflammation, vascular dysfunction and neuronal damage associated with ischemic stroke and vascular cognitive impairment and dementia.

## Figures and Tables

**Figure 1 cells-10-01167-f001:**
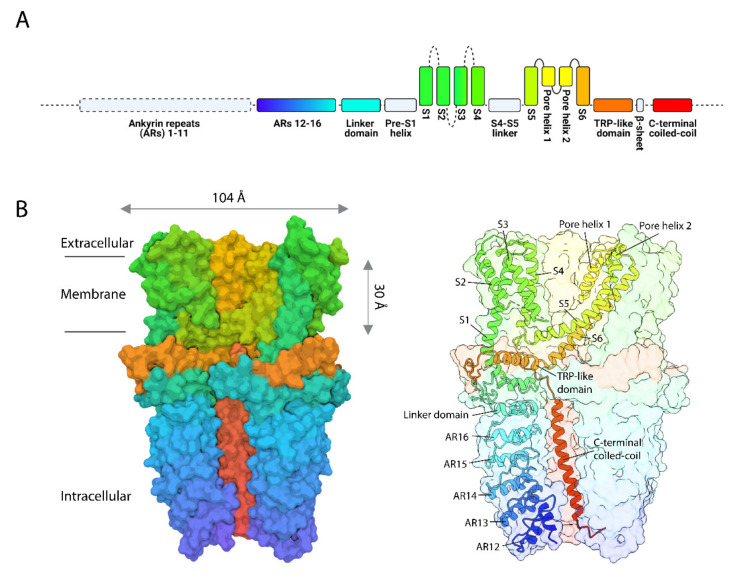
Human TRPA1 structure and 3D reconstruction. (**A**) Major structural domains of hTRPA1 color-coded blue at N-terminus to red at C-terminus. Dashed lines and boxes denote regions in which cryo-EM density was insufficient to resolve structural details, or where definitive assignment of specific residues was not possible. Solid gray boxes indicate structural regions not annotated in (**B**). (**B**) 3D cryo-EM density map of TRPA1 (left; PDB entry 3j9p) and ribbon diagram of a TRPA1 subunit (right) color-matched to linear diagram in (**A**). Figure adapted from [22] (reprinted by permission from Springer Nature: Nature, Copyright 2015) and generated in part with UCSF ChimeraX [40].

**Figure 2 cells-10-01167-f002:**
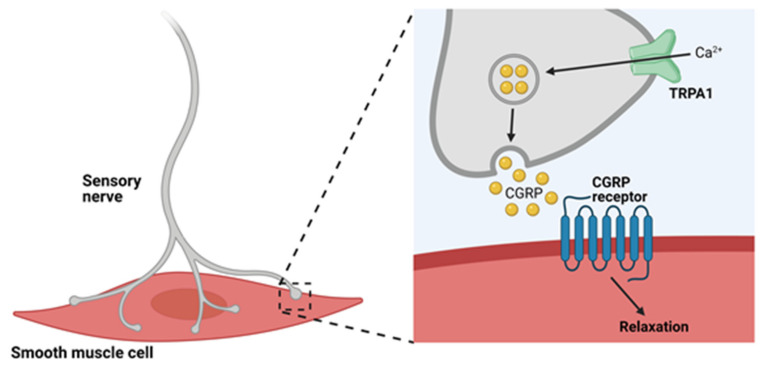
Arterial dilation occurs through activation of TRPA1 in sensory nerves. TRPA1 channel activation on sensory nerves results in Ca^2+^-dependent release of CGRP, which in turn binds to CGRP receptors on the plasma membranes of VSMCs to cause hyperpolarization and relaxation.

**Figure 3 cells-10-01167-f003:**
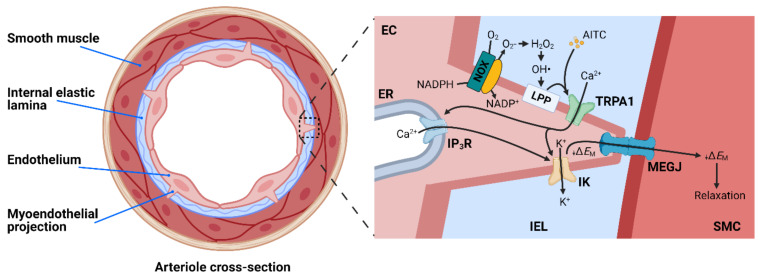
TRPA1 channels cause endothelium-dependent dilation of cerebral arteries. Left: Cross-section of an arteriole. Right: Activation of TRPA1 in the cerebral endothelium by AITC and ROS-derived metabolites: 1) O_2_^−^ generated by NOX enzymes on the endothelial cell plasma membrane is rapidly converted to H_2_O_2_ by superoxide dismutase. 2) In the presence of iron, H_2_O_2_ is converted to OH•, which oxidizes membrane lipids to generate lipid peroxidation products (LPPs), such as 4-HNE. 3) 4-HNE and other LPPs activate TRPA1 channels, and the resulting Ca^2+^ influx is amplified by Ca^2+^-induced Ca^2+^ release through IP_3_Rs on the ER membrane to produce a Ca^2+^ signal that engages Ca^2+^-activated IK channels. 4) K^+^ efflux through IK channels hyperpolarizes the endothelial cell plasma membrane, a signal that is conducted to the underlying smooth muscle by myoendothelial gap junctions (MEGJ) to hyperpolarize the smooth muscle plasma membrane and cause relaxation.

**Figure 4 cells-10-01167-f004:**
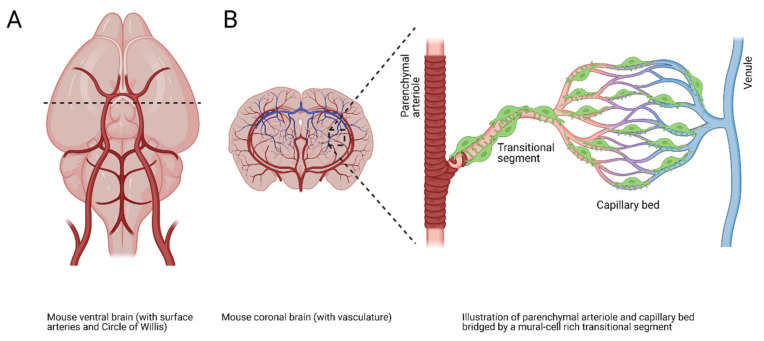
Cerebral vascular overview and microvascular schematic. (**A**) Illustration of the mouse ventral brain with visible surface vasculature. Dotted line indicates coronal section in (**B**). (**B**) Illustration of a mouse coronal brain section showing a complex vascular network composed of a parenchymal arteriole, a mural-cell–ensheathed transitional segment, a capillary bed with pericytes, and an exiting venule.

**Figure 5 cells-10-01167-f005:**
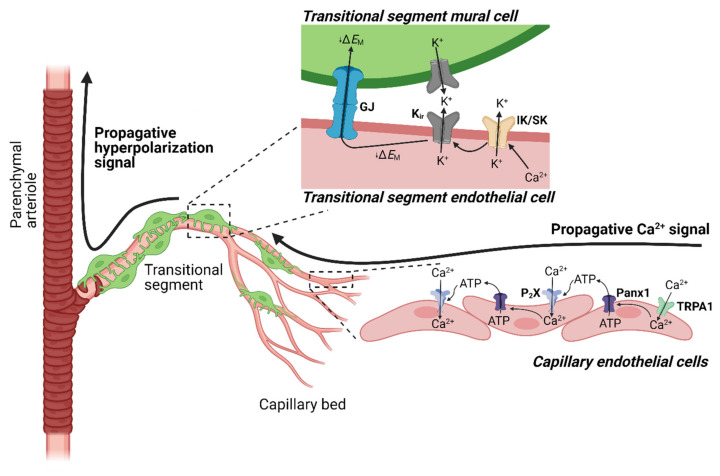
Neurovascular coupling initiated by TRPA1 channels on brain capillary endothelial cells. TRPA1 channel activation on capillary endothelial cells results in Ca^2+^ influx and triggers Ca^2+^ signals that propagate through the capillary network. The release of ATP through Panx1 channels and subsequent activation of purinergic P_2_X receptors sustains this propagating signal. Upon arriving at the post-arteriole transitional segment, the propagating Ca^2+^ signal from the capillary bed is converted into a hyperpolarizing electrical signal by Ca^2+^-activated IK and SK channels. The resulting membrane hyperpolarization is amplified and propagated by K_ir_ channels, resulting in dilation of the upstream parenchymal arteriole and increased blood flow.

**Table 1 cells-10-01167-t001:** List of TRPA1 activators. m-, r-, and hTRPA1 isoforms correspond to mouse, rat, and human TRPA1, respectively).

Agonist	Source	EC_50_; Isoform	Reference
Allyl isothiocyanate (AITC)	Mustard	33 μM; mTRPA1 11 μM; rTRPA1 64 μM; hTRPA1	[17,38,47]
Cinnamaldehyde	Cinnamon	100 µM; mTRPA1	[47]
Allicin	Garlic	1.3 µM; mTRPA1 1.9 µM; hTRPA1	[58,59]
Propofol	Anesthetic	17 μM; mTRPA1	[60]
Lidocaine	Anesthetic	5.7 mM; rTRPA1 24 mM; hTRPA1	[44]
4-HNE	Reactive oxygen species	13–20 µM; mTRPA1 27 µM; rTRPA1 5 µM; hTRPA1	[41,46,61,62]
4-ONE	Reactive oxygen species	1.9 µM; mTRPA1 5.8 µM; hTRPA1	[41,62]
4-HHE	Reactive oxygen species	38.9 µM; mTRPA1 ≥4.3 µM; hTRPA1	[41,62]
H_2_O_2_	Reactive oxygen species	230 μM; mTRPA1	[41]
15-deoxy-delta(12,14)-prostaglandin J(2) [15d-PGJ(2)]	Reactive oxygen species	5.6 μM; mTRPA1	[41]
ASP7663	Synthetic	0.50 µM; mTRPA1 0.54 µM; rTRPA1 0.51 µM; hTRPA1	[63]
JT010	Synthetic	0.65 nM; hTRPA1	[48]
GNE551	Synthetic	254 nM; hTRPA1	[64]
WaTx	Peptidergic toxin	6 nM; mTRPA1 15 nM; rTRPA1, 16nM; hTRPA1	[65].

**Table 2 cells-10-01167-t002:** List of TRPA1 antagonists. m-, r-, and hTRPA1 isoforms correspond to mouse, rat, and human TRPA1, respectively.

Antagonist	IC_50;_ isoform	Reference
HC-030031	7.6 μM; rTRPA1 5.3–6.2 μM; hTRPA1	[68]
Chembridge-5861528	14.3–18.7 µM; hTRPA1	[69,75]
AP-18	4.5 μM; mTRPA1 3.1 μM; hTRPA1	[70,71]
A-967079	289 nM; rTRPA1 67 nM; hTRPA1	[72]
Compound 10	45 nM; rTRPA1 170 nM; hTRPA1	[73]
Compound 31	85 nM; rTRPA1 15 nM; hTRPA1	[74]

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
