# Peer review of "Transient Receptor Potential Channel Ankyrin 1: A Unique Regulator of Vascular Function"

_cells, 2021, doi:10.3390/cells10051167_

Round 1

Reviewer 1 Report

The authors have written an enjoyable and clear review on the function of neural and vascular TRPA1 channels in the regulation of vascular functions. They provide information about the structure, pharmacology, physiological and pathophysiological role of TRPA1 channels.

The role of TRPA1 channel activation in neurogenic sensory vasodilatation is discussed but no data are provided about the density of TRPA1 expressing, CGRP containing afferents innervating vascular elements. Please provide some quantitative data about this fraction of primary sensory neurons, since it helps the estimation of the significance of this mechanism in neurogenic vasodilatation.

A quite important way of TRPA1 activation is missing from the review. Eberhardt et al. proved that the interaction of two endogenous mediators NO and H2S produces nitroxyl that activates TRPA1 receptors and releases CGRP from intracranial nociceptors (Eberhardt et al. H2S and NO cooperatively regulate vascular tone by activating a neuroendocrine HNO–TRPA1–CGRP signalling pathway, NATURE COMMUNICATIONS | 5:4381 | DOI: 10.1038/ncomms5381). Please add this mechanism to the review as an endogenous way of receptor activation.

Minor issues:

Line 108: please correct „allyl isothiocyanate”

Lines 171, 176, 189, 194,198, 202, 203: please correct „CGRP” and „CGRP8-37”

Author Response

We thank the reviewers for their suggestions for our manuscript entitled "Transient receptor potential channel ankyrin 1: A unique regulator of vascular function". We have revised our manuscript to address the reviewer comments and have included our responses below:

  • The role of TRPA1 channel activation in neurogenic sensory vasodilatation is discussed but no data are provided about the density of TRPA1 expressing, CGRP containing afferents innervating vascular elements. Please provide some quantitative data about this fraction of primary sensory neurons, since it helps the estimation of the significance of this mechanism in neurogenic vasodilatation.

Response: Bautista et al. [74] reported that TRPA1 is expressed by a subpopulation of CGRP-containing sensory neurons in a qualitative manner using immunoreactivity in perivascular nerve fibers in mesenteric artery.  However, to our knowledge, there have been no studies performed quantifying the density of TRPA1 expressing, CGRP containing neurons in the vasculature.

  • A quite important way of TRPA1 activation is missing from the review. Eberhardt et al. proved that the interaction of two endogenous mediators NO and H2S produces nitroxyl that activates TRPA1 receptors and releases CGRP from intracranial nociceptors (Eberhardt et al. H2S and NO cooperatively regulate vascular tone by activating a neuroendocrine HNO–TRPA1–CGRP signaling pathway, NATURE COMMUNICATIONS | 5:4381 | DOI: 10.1038/ncomms5381). Please add this mechanism to the review as an endogenous way of receptor activation.

Response: Thank you - we have incorporated this mechanism into the review.

  • Line 108: please correct „allyl isothiocyanate”; Lines 171, 176, 189, 194,198, 202, 203: please correct „CGRP” and „CGRP8-37”

Response: Thank you - we have corrected these typos.

Reviewer 2 Report

Chapters 1-3 are missing any illustrations.
There are detailed mentions of specific amino acids,
or "slang" names for different parts of the TRPA1 channel.
Thus, of course, these introductory chapters can only be understandable for a narrow group of insiders,
which, however, will skip these chapters.
For uninitiated readers, these introductory chapters without adequate illustrations will be an insurmountable barrier,
which would discourage them from reading the following chapters.

It is therefore necessary to adequately illustrate Chapters 1-3.
There is enough source material for these purposes.
The manuscript mentions three studies (ref. 22, 52, 58),
describing cryoEM structures of the TRPA1 ion channel.

And there are several other similar papers - see below.

This TRPA1 structures can be freely downloaded from the www.rcsb.org protein database.
By means of the UCSF Chimera or ICM Molsoft software packages,
it is easy to create illustrations of the TRPA1 channel structure,
binding sites of ligands (agonists as well as antagonists)
and crucial amino acids interacting with them.

22. Paulsen C. E., Armache J.-P., Gao Y., Cheng Y., Julius D.
Structure of the TRPA1 ion channel suggests regulatory mechanisms.
Nature 520 (2015) 511-517
3J9P

52. Suo Y., Wang Z., Zubcevic L., Hsu A.L., He Q., Borgnia M.J., Ji R.-R. and Lee S.-Y.
Structural Insights into Electrophile Irritant Sensing by the Human TRPA1 Channel
Neuron 105 (2020) 882-894 
6PQQ 6PQP 6PQO

58. Liu C. et al.
A Non-covalent Ligand Reveals Biased Agonism of the TRPA1 Ion Channel
Neuron 109 (2021) 1-12
6X2J

??. Zhao J., King J.V.L., Paulsen C.E., Cheng Y., Julius D.
Irritant-evoked activation and calcium modulation of the TRPA1 receptor
Nature 585 (2020) 141-145
6V9X 6V9W 6V9Y 6V9V

??. Terrett J.A. et al.
Tetrahydrofuran-Based Transient Receptor Potential Ankyrin 1 (TRPA1) Antagonists:
Ligand-Based Discovery, Activity in a Rodent Asthma Moldel,
an Mechanism-of-Action via Cryogenic Electron Microscopy
J. Med. Chem. 64 (2021) 3843-3869
7JUP

??. to be publised
6WJP

TRPA1 - www.rcsb.org

6V9X - open state       
6V9W - calcium
6V9Y - A-967079              *** Figure 1 - A96 (closed) vs. iodoacetamide (activated)  
6V9V - bodipy-iodoacetamide
       calcium

6PQQ - apo
6PQP - agonist BITC 
6PQO - agonist JT010         *** Figure 1

3J9P - antagonist A-967079   *** Figure 6, Extended Data Figure 7

6X2J - agonist GNE551        *** Figure 3, Figure 6

7JUP - antagonist compound 21

6WJ5 - antagonist GDC-0334

Author Response

  • Chapters 1-3 are missing any illustrations. There are detailed mentions of specific amino acids, or "slang" names for different parts of the TRPA1 channel. Thus, of course, these introductory chapters can only be understandable for a narrow group of insiders, which, however, will skip these chapters. For uninitiated readers, these introductory chapters without adequate illustrations will be an insurmountable barrier, which would discourage them from reading the following chapters.

It is therefore necessary to adequately illustrate Chapters 1-3. There is enough source material for these purposes. The manuscript mentions three studies (ref. 22, 52, 58), describing cryoEM structures of the TRPA1 ion channel.

And there are several other similar papers - see below.

This TRPA1 structures can be freely downloaded from the www.rcsb.org protein database. By means of the UCSF Chimera or ICM Molsoft software packages, it is easy to create illustrations of the TRPA1 channel structure, binding sites of ligands (agonists as well as antagonists) and crucial amino acids interacting with them.

  1. Paulsen C. E., Armache J.-P., Gao Y., Cheng Y., Julius D.

Structure of the TRPA1 ion channel suggests regulatory mechanisms.

Nature 520 (2015) 511-517

3J9P

  1. Suo Y., Wang Z., Zubcevic L., Hsu A.L., He Q., Borgnia M.J., Ji R.-R. and Lee S.-Y.

Structural Insights into Electrophile Irritant Sensing by the Human TRPA1 Channel

Neuron 105 (2020) 882-894

6PQQ 6PQP 6PQO

  1. Liu C. et al.

A Non-covalent Ligand Reveals Biased Agonism of the TRPA1 Ion Channel

Neuron 109 (2021) 1-12

6X2J

??. Zhao J., King J.V.L., Paulsen C.E., Cheng Y., Julius D.

Irritant-evoked activation and calcium modulation of the TRPA1 receptor

Nature 585 (2020) 141-145

6V9X 6V9W 6V9Y 6V9V

??. Terrett J.A. et al.

Tetrahydrofuran-Based Transient Receptor Potential Ankyrin 1 (TRPA1) Antagonists:

Ligand-Based Discovery, Activity in a Rodent Asthma Moldel, an Mechanism-of-Action via Cryogenic Electron Microscopy

  1. Med. Chem. 64 (2021) 3843-3869

7JUP

??. to be publised

6WJP

TRPA1 - www.rcsb.org

6V9X - open state      

6V9W - calcium

6V9Y - A-967079              *** Figure 1 - A96 (closed) vs. iodoacetamide (activated) 

6V9V - bodipy-iodoacetamide

       calcium

6PQQ - apo

6PQP - agonist BITC

6PQO - agonist JT010         *** Figure 1

3J9P - antagonist A-967079   *** Figure 6, Extended Data Figure 7

6X2J - agonist GNE551        *** Figure 3, Figure 6

7JUP - antagonist compound 21

6WJ5 - antagonist GDC-0334

Response: Thank you - we have incorporated a structural illustration of TRPA1 adapted from Paulsen et al. [22] into the review.

Round 2

Reviewer 1 Report

The questions have been answered satisfactorily.

Reviewer 2 Report

Accept in present form